# Approximation Ratios of Graph Neural Networks for Combinatorial Problems

**Ryoma Sato**[1,2]     **Makoto Yamada**[1,2,3]     **Hisashi Kashima**[1,2]
[1]Kyoto University     [2]RIKEN AIP     [3]JST PRESTO
{r.sato@ml.ist.i, myamada@i, kashima@i}.kyoto-u.ac.jp

## Abstract

In this paper, from a theoretical perspective, we study how powerful graph neural networks (GNNs) can be for learning approximation algorithms for combinatorial problems. To this end, we first establish a new class of GNNs that can solve a strictly wider variety of problems than existing GNNs. Then, we bridge the gap between GNN theory and the theory of distributed local algorithms. We theoretically demonstrate that the most powerful GNN can learn approximation algorithms for the minimum dominating set problem and the minimum vertex cover problem with some approximation ratios with the aid of the theory of distributed local algorithms. We also show that most of the existing GNNs such as GIN, GAT, GCN, and GraphSAGE cannot perform better than with these ratios. This paper is the first to elucidate approximation ratios of GNNs for combinatorial problems. Furthermore, we prove that adding coloring or weak-coloring to each node feature improves these approximation ratios. This indicates that preprocessing and feature engineering theoretically strengthen model capabilities.

## 1 Introduction

Graph neural networks (GNNs) [8, 9, 12, 22] is a novel machine learning method for graph structures. GNNs have achieved state-of-the-art performance in various tasks, including chemo-informatics [7], question answering systems [23], and recommendation systems [31], to name a few.

Recently, machine learning methods have been applied to combinatorial problems [4, 11, 16, 27] to automatically obtain novel and efficient algorithms. Xu et al. [30] analyzed the capability of GNNs for solving the graph isomorphism problem, and they found that GNNs cannot solve it but they are as powerful as the Weisfeiler-Lehman graph isomorphism test.

The minimum dominating set problem, minimum vertex cover problem, and maximum matching problem are examples of important combinatorial problems other than the graph isomorphism problem. These problems are all NP-hard. Therefore, under the assumption that $P \neq NP$, GNNs cannot exactly solve these problems because they run in polynomial time with respect to input size. For NP-hard problems, many approximation algorithms have been proposed to obtain sub-optimal solutions in polynomial time [25], and approximation ratios of these algorithms have been studied to guarantee the performance of these algorithms.

In this paper, we study the approximation ratios of algorithms that GNNs can learn for combinatorial problems. To analyze the approximation ratios of GNNs, we bridge the gap between GNN theory and the theory of distributed local algorithms. Here, distributed local algorithms are distributed algorithms that use only a constant number of synchronous communication rounds [1, 10, 24]. Thanks to their relationship with distributed local algorithms, we can elucidate the lower bound of the approximation ratios of algorithms that GNNs can learn for combinatorial problems. As an example of our results, if the input feature of each node is the node degree alone, no GNN can solve $(\Delta + 1 - \varepsilon)$-approximation

for the minimum dominating set problem or $(2 - \varepsilon)$-approximation for the minimum vertex cover problem, where $\varepsilon > 0$ is any real number and $\Delta$ is the maximum node degree.

In addition, thanks to this relationship, we find vector-vector consistent GNNs ($\text{VV}_\text{C}$-GNNs), which are a novel class of GNNs. $\text{VV}_\text{C}$-GNNs have strictly stronger capability than existing GNNs and have the same capability as a computational model of distributed local algorithms. Based on our key finding, we propose the consistent port numbering GNNs (CPNGNNs), which is the most powerful GNN model among $\text{VV}_\text{C}$-GNNs. That is, for any graph problem that a $\text{VV}_\text{C}$-GNN can solve, there exists a parameter of CPNGNNs that can also solve it. Interestingly, CPNGNNs are strictly more powerful than graph isomorphism networks (GIN), which were considered to be the most powerful GNNs [30]. Furthermore, CPNGNNs achieve optimal approximation ratios among GNNs: CPNGNNs can solve $(\Delta + 1)$-approximation for the minimum dominating set problem and 2-approximation for the minimum vertex cover problem.

However, these approximation ratios are unsatisfactory because they are as high as those of simple greedy algorithms. One of the reasons for these high approximation ratios is that we only use node degrees as node features. We show that adding coloring or weak coloring to each node feature strengthens the capability of GNNs. For example, if we use weak 2-coloring as a node feature in addition to node degree, CPNGNNs can solve $\left(\frac{\Delta+1}{2}\right)$-approximation for the minimum dominating set problem. Considering that any graph has weak 2-coloring and that we can easily calculate weak 2-coloring in linear time, it is interesting that such preprocessing and feature engineering can theoretically strengthen the model capability.

The contributions of this paper are summarized as follows:

- We reveal the relationships between the theory of GNNs and distributed local algorithms. Namely, we show that the set of graph problems that GNN classes can solve is the same as the set of graph problems that distributed local algorithm classes can solve.

- We propose CPNGNNs, which is the most powerful GNN among the proposed GNN class.

- We elucidate the approximation ratios of GNNs for combinatorial problems including the minimum dominating set problem and the minimum vertex cover problem. This is the first paper to elucidate the approximation ratios of GNNs for combinatorial problems.

## 2 Related Work

### 2.1 Graph Neural Networks

GNNs were first introduced by Gori et al. [8] and Scarselli et al. [22]. They obtained the node embedding by recursively applying the propagation function until convergence. Recently, Kipf and Welling [12] proposed graph convolutional networks (GCN), which significantly outperformed existing methods, including non-neural network-based approaches. Since then, many graph neural networks have been proposed, such as GraphSAGE [9] and the graph attention networks (GATs) [26].

Vinyals et al. [27] proposed pointer networks, which can solve combinatorial problems on a plane, such as the convex hull problem and the traveling salesman problem. Bello et al. [4] trained pointer networks using reinforcement learning to automatically obtain novel algorithms for these problems. Note that pointer networks are not GNNs. However, we introduce them here because they were the first to solve combinatorial problems using deep learning. Khalil et al. [11] and Li et al. [16] used GNNs to solve combinatorial problems. They utilized search methods with GNNs, whereas we use only GNNs to focus on the capability of GNNs.

Xu et al. [30] analyzed the capability of GNNs. They showed that GNNs cannot solve the graph isomorphism problem and that the capability of GNNs is at most the same as that of the Weisfeiler-Lehman graph isomorphism test. They also proposed the graph isomorphism networks (GIN), which are as powerful as the Weisfeiler-Lehman graph isomorphism test. Therefore, the GIN is the most powerful GNNs. The motivation of this paper is the same as that of Xu et al.'s work [30] but we consider not only the graph isomorphism problem but also the minimum dominating set problem, minimum vertex cover problem, and maximum matching problem. Furthermore, we find the approximation ratios of these problems for the first time and propose GNNs more powerful than GIN.

---

**Algorithm 1** Calculating the embedding of a node using GNNs

---

**Require:** Graph $G = (V, E, \boldsymbol{X})$; Parameters $\boldsymbol{\theta}$; Aggregation function $f_{\boldsymbol{\theta}}^{(l)} (l = 1, \ldots, L)$.
**Ensure:** Embedding of nodes $\boldsymbol{z} \in \mathbb{R}^{n \times d_{L+1}}$
  1: $\boldsymbol{z}_v^{(1)} \leftarrow \boldsymbol{x}_v \ (\forall v \in V)$
  2: **for** $l = 1, \ldots, L$ **do**
  3:    **for** $v \in V$ **do**
  4:       $\boldsymbol{z}_v^{(l+1)} \leftarrow f_{\boldsymbol{\theta}}^{(l)}$ (aggregated information from neighbor nodes of $v$)
  5:    **end for**
  6: **end for**
  7: **return** $\boldsymbol{z}^{(L+1)}$

---

## 2.2 Distributed Local Algorithms

A distributed local algorithm is a distributed algorithm that runs in constant time. More specifically, in a distributed local algorithm, we assume each node has infinite computational resources and decides the output within a constant number of communication rounds with neighboring nodes. For example, distributed local algorithms are used for controlling wireless sensor networks [13], constructing self-stabilization algorithms [14, 18], and building sublinear-time algorithms [20].

Distributed local algorithms were first studied by Angluin [1], Linial [17], and Naor and Stockmeyer [18]. Angluin [1] showed that deterministic distributed algorithms cannot find a center of a graph without any unique node identifiers. Linial [17] showed that no distributed *local* algorithms can solve 3-coloring of cycles, and they require $\Omega(\log^* n)$ communication rounds for distributed algorithms to solve the problem. Naor and Stockmeyer [18] showed positive results for distributed local algorithms for the first time. For example, distributed local algorithms can find weak 2-coloring and solve a variant of the dining philosophers problem. Later, several non-trivial distributed local algorithms were found, including 2-approximation for the minimum vertex cover problem [2].

There are many computational models of distributed local algorithms. Some computational models use unique identifiers of nodes [18], port numbering [1], and randomness [19, 28], and other models do not [10]. Furthermore, some results use the following assumptions about the input: degrees are bounded [2], degrees are odd [18], graphs are planar [6], and graphs are bipartite [3]. In this paper, we do not use any unique identifiers nor randomness, but we do use port numbering, and we assume the degrees are bounded. We describe our assumptions in detail in Section 3.1.

# 3 Preliminaries

## 3.1 Problem Setting

Here, we first describe the notation used in this paper and then we formulate the graph problem.

**Notation.** For a positive integer $k \in \mathbb{Z}^+$, let $[k]$ be the set $\{1, 2, \ldots, k\}$. Let $G = (V, E, \boldsymbol{X})$ be a input graph, where $V$ is a set of nodes, $E$ is a set of edges, and $\boldsymbol{X} \in \mathbb{R}^{|V| \times d_0}$ is a feature matrix. We represent an edge of a graph $G = (V, E, \boldsymbol{X})$ as an unordered pair $\{u, v\}$ with $u, v \in V$. We write $n = |V|$ for the number of nodes and $m = |E|$ for the number of edges. The nodes $V$ are considered to be numbered with $[n]$. (i.e., we assume $V = [n]$.) For a node $v \in V$, $\deg(u)$ denotes the degree of node $v$ and $\mathcal{N}(v)$ denotes the set of neighbors of node $v$.

A GNN model $N_{\boldsymbol{\theta}}(G, v)$ is a function parameterized by $\boldsymbol{\theta}$ that takes a graph $G$ and a node $v \in V$ as input and output the label $y_v \in Y$ of node $v$, where $Y$ is a set of labels. We study the expression capability of the function family $N_{\boldsymbol{\theta}}$ for combinatorial graph problems with the following assumptions.

**Assumption 1 (Bounded-Degree Graphs).** In this paper, we consider only bounded-degree graphs. In other words, for a fixed (but arbitrary) constant $\Delta$, we assume that the degree of each node of the input graphs is at most $\Delta$. This assumption is natural because there are many bounded-degree graphs in the real world. For example, degrees in molecular graphs are bounded by four, and the degrees in computer networks are bounded by the number of LAN ports of routers. Moreover, the

bounded-degree assumption is often used in distributed local algorithms [17, 18, 24]. For each positive integer $\Delta \in \mathbb{Z}^+$, let $\mathcal{F}(\Delta)$ be the set of all graphs with maximum degrees of $\Delta$ at most.

**Assumption 2 (Node Features).** We do not consider node features other than those that can be derived from the input graph itself for focusing on graph theoretic properties. When there are no node features available, the degrees of nodes are sometimes used [9, 21, 30]. Therefore, we use only the degree of a node as the node feature (i.e., $z_v^{(1)} = \text{ONEHOT}(\deg(v))$) unless specified. Later, we show that using coloring or weak coloring of the input graph in addition to degrees of nodes as node features makes models theoretically more powerful.

**Graph Problems.** A *graph problem* is a function $\Pi$ that associates a set $\Pi(G)$ of *solutions* with each graph $G = (V, E)$. Each solution $S \in \Pi(G)$ is a function $S \colon V \to Y$. $Y$ is a finite set that is independent of $G$. We say a GNN model $N_{\boldsymbol{\theta}}$ *solves* a graph problem $\Pi$ if for any $\Delta \in \mathbb{Z}^+$, there exists a parameter $\boldsymbol{\theta}$ such that for any graph $G \in \mathcal{F}(\Delta)$, $N_{\boldsymbol{\theta}}(G, \cdot)$ is in $\Pi(G)$. For example, let $Y$ be a set of labels of nodes, let $L(G) \colon V \to Y$ be the ground truth of a multi-label classification problem for a graph $G$ (*i.e.,* $L(G)(v)$ denotes the ground truth label of node $v \in V$), and let $\Pi(G) = \{f \colon V \to \{0, 1\} \mid |\{v \in V \mid f(v) = L(G)(v)\}| \geq 0.9 \cdot |V|\}$. This graph problem $\Pi$ corresponds to a multi-label classification problem. A GNN model $N_{\boldsymbol{\theta}}$ solves $\Pi$ means there exists a parameter $\boldsymbol{\theta}$ of the model such that achieves an accuracy 0.9 for this problem. Other examples of graph problems are combinatorial problems. Let $C(G) \subset V$ be the minimum vertex cover of a graph $G$, let $Y = \{0, 1\}$, and let $\Pi(G) = \{f \colon V \to \{0, 1\} \mid D = \{v \mid f(v) = 1\} \text{ is a vertex cover and } |D| \leq 2 \cdot |C(G)|\}$. This graph problem $\Pi$ corresponds to 2-approximation for the minimum vertex cover problem.

## 3.2 Known Model Classes

We introduce two known classes of GNNs, which include GraphSAGE [9], GCN [12], GAT [26], and GIN [30].

**MB-GNNs.** A layer of an existing GNN can be written as

$$z_v^{(l+1)} = f_{\boldsymbol{\theta}}^{(l)}(z_v^{(l)}, \text{MULTISET}(z_u^{(l)} \mid u \in \mathcal{N}(v))),$$

where $f_{\boldsymbol{\theta}}^{(l)}$ is a learnable aggregation function. We call GNNs that can be written in this form multiset-broadcasting GNNs (MB-GNNs) — multiset because they aggregate features from neighbors as a multiset and broadcasting because for any $v \in \mathcal{N}(u)$, the "message" [7] from $u$ to $v$ is the same (i.e., $z_u$). GraphSAGE-mean [9] is an example of MB-GNNs because a layer of GraphSAGE-mean is represented by the following equation:

$$z_v^{(l+1)} = \text{CONCAT}(z_v^{(l)}, \frac{1}{|\mathcal{N}(v)|} \sum_{u \in \mathcal{N}(v)} W^{(l)} z_u^{(l)}),$$

where CONCAT concatenates vectors into one vector. Other examples of MB-GNNs are GCN [12], GAT [26], and GIN [30].

**SB-GNNs.** The another existing class of GNNs in the literature is set-broadcasting GNNs (SB-GNNs), which can be written as the following form:

$$z_v^{(l+1)} = f_{\boldsymbol{\theta}}^{(l)}(z_v^{(l)}, \text{SET}(z_u^{(l)} \mid u \in \mathcal{N}(v))).$$

GraphSAGE-pool [9] is an example of SB-GNNs because a layer of GraphSAGE-mean is represented by the following equation:

$$z_v^{(l+1)} = \max(\{\sigma(W^{(l)} z_u^{(l)} + b^{(l)}) \mid u \in \mathcal{N}(v)\}).$$

Clearly, SB-GNNs are a subclass of MB-GNNs. Xu et al. [30] discussed the differences in capability of SB-GNNs and MB-GNNs. We show that MB-GNNs are strictly stronger than SB-GNNs in another way in this paper.

## 4 Novel Class of GNNs

In this section, we first introduce a GNN class that is more powerful than MB-GNNs and SB-GNNs. To make GNN models more powerful than MB-GNNs, we introduce the concept of port numbering [1, 10] to GNNs.

**Port Numbering.** A port of a graph $G$ is a pair $(v, i)$, where $v \in V$ and $i \in [\deg(v)]$. Let $P(G) = \{(v, i) \mid v \in V, i \in [\deg(v)]\}$ be the set of all ports of a graph $G$. A port numbering of a graph $G$ is the function $p \colon P(G) \to P(G)$ such that for any edge $\{u, v\}$, there exist $i \in [\deg(u)]$ and $j \in [\deg(v)]$ such that $p(u, i) = (v, j)$. We say that a port numbering is *consistent* if $p$ is an involution (i.e., $\forall (v, i) \in P(G)\ p(p(v, i)) = (v, i)$). We define the functions $p_{\text{tail}} \colon V \times \Delta \to V \cup \{-\}$ and $p_{\text{n}} \colon V \times \Delta \to \Delta \cup \{-\}$ as follows:

$$p_{\text{tail}}(v, i) = \begin{cases} u \in V \ (\exists j \in [\deg(u)]\ s.t.\ p(u, j) = (v, i)) & (i \leq \deg(v)) \\ - & (\text{otherwise}), \end{cases}$$

$$p_{\text{n}}(v, i) = \begin{cases} j \in [\deg(p_{\text{tail}}(v, i))]\ (p(p_{\text{tail}}(v, i), j) = (v, i)) & (i \leq \deg(v)) \\ - & (\text{otherwise}), \end{cases}$$

where $-$ is a special symbol that denotes the index being out of range. Note that these functions are well-defined because there always exists only one $u \in V$ for $p_{\text{tail}}$ and $j \in [\deg(p_{\text{tail}}(v, i))]$ for $p_{\text{n}}$ if $i \leq \deg(v)$. Intuitively, $p_{\text{tail}}(v, i)$ represents the node that sends messages to the port $i$ of node $v$ and $p_{\text{n}}(v, i)$ represents the port number of the node $p_{\text{tail}}(v, i)$ that sends messages to the port $i$ of node $v$.

The GNN class we introduce in the following uses a consistent port numbering to calculate embeddings. Intuitively, SB-GNNs and MB-GNNs send the same message to all neighboring nodes. GNNs can send different messages to neighboring nodes by using port numbering, and this strengthens model capability.

**VV$_\text{C}$-GNNs.** Vector-vector consistent GNNs (VV$_\text{C}$-GNNs) are a novel class of GNNs that we introduce in this paper. They calculate an embedding with the following formula:

$$\boldsymbol{z}_v^{(l+1)} = f_{\boldsymbol{\theta}}^{(l)}(\boldsymbol{z}_v^{(l)}, \boldsymbol{z}_{p_{\text{tail}}(v,1)}^{(l)}, p_{\text{n}}(v, 1), \boldsymbol{z}_{p_{\text{tail}}(v,2)}^{(l)}, p_{\text{n}}(v, 2), \ldots, \boldsymbol{z}_{p_{\text{tail}}(v,\Delta)}^{(l)}, p_{\text{n}}(v, \Delta)).$$

If the index of $\boldsymbol{z}$ is the special symbol $-$, we also define the embedding as the special symbol $-$ (i.e., $\boldsymbol{z}_- = -$). To calculate embeddings of nodes of a graph $G$ using a GNN with port numbering, we first calculate one consistent port numbering $p$ of $G$, and then we input $G$ and $p$ to the GNN. Note that we can calculate a consistent port numbering of a graph in linear time by numbering edges one by one. We say a GNN class $\mathcal{N}$ with port numbering *solves* a graph problem $\Pi$ if for any $\Delta \in \mathbb{Z}^+$, there exists a GNN $N_{\boldsymbol{\theta}} \in \mathcal{N}$ and its parameter $\boldsymbol{\theta}$ such that for any graph $G \in \mathcal{F}(\Delta)$, for *any* consistent port numbering $p$ of $G$, the output $N_{\boldsymbol{\theta}}(G, p, \cdot)$ is in $\Pi(G)$. We show that using port numbering theoretically improves model capability in Section 5.2. We propose CPNGNNs, an example of VV$_\text{C}$-GNNs, in Section 6.

## 5 GNNs with Distributed Local Algorithms

In this section, we discuss the relationship between GNNs and distributed local algorithms. Thanks to this relationship, we can elucidate the theoretical properties of GNNs.

### 5.1 Relationship with Distributed Local Algorithms

A distributed local algorithm is a distributed algorithm that runs in constant time. More specifically, in a distributed local algorithm, we assume each node has infinite computational resources and decides the output within a constant number of communication rounds with neighboring nodes. In this paper, we show a clear relationship between distributed local algorithms and GNNs for the first time.

There are several well-known models of distributed local algorithms [10]. Namely, in this paper, we introduce the SB(1), MB(1), and VV$_\text{C}$(1) models. As their names suggest, they correspond to SB-GNNs, MB-GNNs, and VV$_\text{C}$-GNNs, respectively.

**Assumption 3 (Finite Node Features):** The number of possible node features is finite.

Assumption 3 restricts node features be discrete. However, Assumption 3 does include the node degree feature ($\in [\Delta]$) and node coloring feature ($\in \{0, 1\}$).

**Theorem 1.** *Let $\mathcal{L}$ be SB, MB, or VV$_\text{C}$. Under Assumption 3, the set of graph problems that at least one $\mathcal{L}$-GNN can solve is the same as the set of graph problems that at least one distributed local algorithm on the $\mathcal{L}(1)$ model solve.*

**Algorithm 2** CPNGNN: The most powerful VV_C-GNN

---

**Require:** Graph $G = (V, E, \boldsymbol{X})$; Maximum degree $\Delta \in \mathbb{Z}^+$; Weight matrix $\boldsymbol{W}^{(l)} \in \mathbb{R}^{d_{l+1} \times (d_l + \Delta(d_l + 1))}(l = 1, \ldots, L)$.
**Ensure:** Output for the graph problem $\boldsymbol{y} \in Y^n$
 1: calculate a consistent port numbering $p$
 2: $\boldsymbol{z}_v^{(1)} \leftarrow \boldsymbol{x}_v \ (\forall v \in V)$
 3: **for** $l = 1, \ldots, L$ **do**
 4:  **for** $v \in V$ **do**
 5:   $\boldsymbol{z}_v^{(l+1)} \leftarrow \boldsymbol{W}^{(l)} \text{CONCAT}(\boldsymbol{z}_v^{(l)}, \boldsymbol{z}_{p_{\text{tail}}(v,1)}^{(l)}, p_{\text{n}}(v,1), \boldsymbol{z}_{p_{\text{tail}}(v,2)}^{(l)}, p_{\text{n}}(v,2), \ldots, \boldsymbol{z}_{p_{\text{tail}}(v,\Delta)}^{(l)}, p_{\text{n}}(v,\Delta))$

 6:   $\boldsymbol{z}_v^{(l+1)} \leftarrow \text{RELU}(\boldsymbol{z}_v^{(l+1)})$
 7:  **end for**
 8: **end for**
 9: **for** $v \in V$ **do**
10:  $\boldsymbol{z}_v \leftarrow \text{MULTILAYERPERCEPTRON}(\boldsymbol{z}_v^{(L+1)})$   # calculate the final embedding of a node $v$.
11:  $\boldsymbol{y}_v \leftarrow \text{argmax}_{i \in [d_{L+1}]} \boldsymbol{z}_{vi}$   # output the index of the maximum element.
12: **end for**
13: **return** $\boldsymbol{y}$

---

All proofs are available in the supplementary materials. In fact, the following stronger properties hold: (i) any $\mathcal{L}$-GNN can be simulated by the $\mathcal{L}(1)$ model and (ii) any distributed local algorithm on $\mathcal{L}(1)$ model can be simulated by an $\mathcal{L}$-GNN. The former is obvious because GNNs communicate with neighboring nodes in $L$ rounds, where $L$ is the number of layers. The latter is natural because the definition of $\mathcal{L}$-GNNs (Section 3.2 and 4) is intrinsically the same as the definition of the $\mathcal{L}(1)$ model. Thanks to Theorem 1, we can prove which combinatorial problems GNNs can/cannot solve by using theoretical results on distributed local algorithms.

### 5.2 Hierarchy of GNNs

There are obvious inclusion relations among classes of GNNs. Namely, SB-GNNs are a subclass of MB-GNNs, and MB-GNNs are a subclass of VV_C-GNNs. If a model class $\mathcal{A}$ is a subset of a model class $\mathcal{B}$, the graph problems that $\mathcal{A}$ solves is a subset of the graph problems that $\mathcal{B}$ solves. However, it is not obvious whether the proper inclusion property holds or not. Let $\mathcal{P}_{\text{SB-GNNs}}$, $\mathcal{P}_{\text{MB-GNNs}}$, and $\mathcal{P}_{\text{VV_C-GNNs}}$ be the sets of graph problems that SB-GNNs, MB-GNNs, and VV_C-GNNs can solve *only with the degree features*, respectively. Thanks to the relationship between GNNs and distributed local algorithms, we can show that the proper inclusion properties of these classes hold.

**Theorem 2.** $\mathcal{P}_{\text{SB-GNNs}} \subsetneq \mathcal{P}_{\text{MB-GNNs}} \subsetneq \mathcal{P}_{\text{VV_C-GNNs}}$.

An example graph problem that MB-GNNs cannot solve but VV_C-GNNs can solve is the finding single leaf problem [10]. The input graphs of the problem are star graphs and the ground truth contains only a single leaf node. MB-GNNs cannot solve this problem because for each layer, the embeddings of the leaf nodes are exactly same, and the GNN cannot distinguish these nodes. Therefore, if a GNN includes one leaf node in the output, the other leaf nodes are also included to the output. On the other hand, VV_C-GNNs can distinguish each leaf node using port numbering and can appropriately output only a single node. We confirm this fact through experiments in the supplementary materials.

## 6 Most Powerful GNN for Combinatorial Problems

### 6.1 Consistent Port Numbering Graph Neural Networks (CPNGNNs)

In this section, we propose the most powerful VV_C-GNNs, CPNGNNs. The most similar algorithm to CPNGNNs is GraphSAGE [9]. The key differences between GraphSAGE and CPNGNNs are as follows: (i) CPNGNNs use port numbering and (ii) GPNGNNs aggregate features of neighbors by concatenation. We show pseudo code of CPNGNNs in Algorithm 2. Though CPNGNNs are simple, they are the most powerful among VV_C-GNNs. This claim is supported by Theorem 3, where we do not limit node features to the node degree feature.

**Theorem 3.** *Let $\mathcal{P}_{\text{CPNGNNs}}$ be the set of graph problems that CPNGNNs can solve and $\mathcal{P}_{\text{VV}_C\text{-GNNs}}$ be the set of graph problems that VV$_C$-GNNs can solve. Then, under Appsumtion 3, $\mathcal{P}_{\text{CPNGNNs}} = \mathcal{P}_{\text{VV}_C\text{-GNNs}}$.*

The advantages of CPNGNNs are twofold: they can solve a strictly wider set of graph problems than existing models (Theorem 2 and 3). There are many distributed local algorithms that can be simulated by CPNGNNs and we can prove that CPNGNNs can solve a variety of combinatorial problems (see Section 6.2).

## 6.2 Combinatorial Problems that CPNGNNs Can/Cannot Solve

In Section 5.2, we found that there exist graph problems that certain GNNs can solve but others cannot. However, there remains a question. What kind of graph problems can/cannot GNNs solve? In this paper, we study combinatorial problems, including the minimum dominating set problem, maximum matching problem, and minimum vertex cover problem. If GNNs can solve combinatorial problems, we may automatically obtain new algorithms for combinatorial problems by simply training GNNs. Note that from Theorems 2 and 3, if CPNGNNs cannot solve a graph problem, other GNNs cannot solve the problem. Therefore, it is important to investigate the capability of GPNGNNs to study the limitations of GNNs.

**Minimum Dominating Set Problem.** First, we investigate the minimum dominating set problem.

**Theorem 4.** *The optimal approximation ratio of CPNGNNs for the minimum dominating set problem is $(\Delta + 1)$. In other words, CPNGNNs can solve $(\Delta + 1)$-approximation for the minimum dominating set problem, but for any $1 \leq \alpha < \Delta + 1$, CPNGNNs cannot solve $\alpha$-approximation for the minimum dominating set problem.*

Here, CPNGNNs can solve $f(\Delta)$ approximation for the minimum dominating set problem means that for all $\Delta \in \mathbb{Z}^+$, there exists a paramter $\boldsymbol{\theta}$ such that for all input $G \in \mathcal{F}(\Delta)$, $\{v \in V \mid \text{CPNGNN}_{\boldsymbol{\theta}}(G, v) = 1\}$ forms $f(\Delta)$ approximatoin of the minimum dominating set of $G$. However, $(\Delta + 1)$-approximation is trivial because it can be achieved by outputting all the nodes. Therefore, Theorem 4 says that any GNN is as bad as the trivial algorithm in the worst case, which is unsatisfactory. This is possibly because we only use the degree information of local nodes, and we may improve the approximation ratio if we use information other than node degree. Interestingly, we can improve the approximation ratio just by using weak 2-coloring as a feature of nodes. A weak 2-coloring is a function $c \colon V \to \{0, 1\}$ such that for any node $v \in V$, there exists a neighbor $u \in \mathcal{N}(v)$ such that $c(v) \neq c(u)$. Note that any graph has a weak 2-coloring and that we can calculate a weak 2-coloring in linear time by a breadth-first search. In the theorems below, we use not only the degree $\deg(v)$ but also the color $c(v)$ as a feature vector of a node $v \in V$. There may be many weak 2-colorings of a graph $G$. However, the choice of $c$ is arbitrary.

**Theorem 5.** *If the feature vector of a node is consisted of the degree and the color of a weak 2-coloring, the optimal approximation ratio of CPNGNNs for the minimum dominating set problem is $(\frac{\Delta+1}{2})$. In other words, CPNGNN can solve $(\frac{\Delta+1}{2})$-approximation for the minimum dominating set problem, and for any $1 \leq \alpha < \frac{\Delta+1}{2}$, CPNGNN cannot solve $\alpha$-approximation for the minimum dominating set problem.*

In the minimum dominating set problem, we cannot improve the approximation ratio by using 2-coloring instead of weak 2-coloring.

**Theorem 6.** *Even if the feature vector of a node is consisted of the degree and the color of a 2-coloring, for any $1 \leq \alpha < \frac{\Delta+1}{2}$, CPNGNNs cannot solve $\alpha$-approximation for the minimum dominating set problem.*

**Minimum Vertex Cover Problem.** Next, we investigate the minimum vertex cover problem.

**Theorem 7.** *The optimal approximation ratio of CPNGNNs for the minimum vertex cover problem is $2$. In other words, CPNGNNs can solve $2$-approximation for the minimum vertex cover problem, and for any $1 \leq \alpha < 2$, CPNGNNs cannot solve $\alpha$-approximation for the minimum vertex cover problem.*

The simple greedy algorithm can solve 2-approximation for the minimum vertex cover problem. However, this result is not trivial because the algorithm that GNNs learn is not a regular algorithm but

a distributed local algorithm. The distributed local algorithm for 2-approximation for the minimum vertex cover problem is known but not so simple [2]. This result also says that if one wants to find an approximation algorithm using a machine learning approach with better performance than 2-approximation, they must use a non-GNN model or combine GNNs with other methods (e.g., a search method).

**Maximum Matching Problem.** Lastly, we investigate the maximum matching problem. So far, we have only investigated problems on nodes, not edges. We must specify how GNNs output edge labels. Graph edge problems are defined similarly to graph problems, but their solutionas are functions $E \to Y$. In this paper, we only consider $Y = \{0, 1\}$ and we only use VV$_C$-GNNs for solving graph edge problems. Let $G \in \mathcal{F}(\Delta)$ be a graph and $p$ be a port numbering of $G$. To solve graph edge problems, GNNs output a vector $y(v) \in \{0, 1\}^\Delta$ for each node $v \in V$. For each edge $\{u, v\}$, GNNs include the edge $\{u, v\}$ in the output if and only if $y(u)_i = y(v)_j = 1$, where $p(u, i) = (v, j)$ and $p(v, j) = (u, i)$. Intuitively, each node outputs "yes" or "no" to each incident edge (i.e., a port) and we include an edge in the output if both ends output "yes" to the edge. As with graph problems, we say a class $\mathcal{N}$ of GNNs solves a graph edge problem $\Pi$ if for any $\Delta \in \mathbb{Z}^+$, there exists a GNN $N_{\boldsymbol{\theta}} \in \mathcal{N}$ and its parameter $\boldsymbol{\theta}$ such that for any graph $G \in \mathcal{F}(\Delta)$ and any consistent port numbering $p$ of $G$, the output $N_{\boldsymbol{\theta}}(G, p)$ is in $\Pi(G)$.

We investigate the maximum matching problem in detail. In fact, GNNs cannot solve the maximum matching problem at all.

**Theorem 8.** *For any $\alpha \in \mathbb{R}^+$, CPNGNNs that cannot solve $\alpha$-approximation for the maximum matching problem.*

However, CPNGNNs can approximate the maximum matching problem with weak 2-coloring feature.

**Theorem 9.** *If the feature vector of a node is consisted of the degree and the color of a weak 2-coloring, the optimal approximation ratio of CPNGNNs for the maximum matching problem is $(\frac{\Delta+1}{2})$. In other words, CPNGNNs can solve $(\frac{\Delta+1}{2})$-approximation for the maximum matching problem, and for any $1 \le \alpha < \frac{\Delta+1}{2}$, CPNGNNs cannot solve $\alpha$-approximation for the maximum matching problem.*

Furthermore, if we use 2-coloring instead of weak 2-coloring, we can improve the approximation ratio. In fact, it can achieve any approximation ratio. Note that only a bipartite graph has 2-coloring. Therefore, the graph class is implicitly restricted to bipartite graphs in this case.

**Theorem 10.** *If the feature vector of a node is consisted of the degree and the color of a 2-coloring, for any $1 < \alpha$, CPNGNNs can solve $\alpha$-approximation for the maximum matching problem.*

In this paper, we consider only bounded-degree graphs. This assumption is natural, but it is also important to consider graphs without degree bounds. Dealing with such graphs is difficult because graph problems on them are not constant size [24]. Note that solving graph problems becomes more difficult if we do not have the bounded-degree assumption. Therefore, GNNs cannot solve $(\Delta + 1 - \varepsilon)$-approximation for the minimum dominating set problems or $(2 - \varepsilon)$-approximation for the minimum vertex cover problem in the general case.

# 7   Conclusion

In this paper, we introduced VV$_C$-GNNs, which are a new class of GNNs, and CPNGNNs, which are an example of VV$_C$-GNNs. We showed that VV$_C$-GNNs have the same ability to solve graph problems as a computational model of distributed local algorithms. With the aid of distributed local algorithm theory, we elucidated the approximation ratios of algorithms that CPNGNNs can learn for combinatorial graph problems such as the minimum dominating set problem and the minimum vertex cover problem. This paper is the first to show the approximation ratios of GNNs for combinatorial problems. Moreover, this is a lower bound of approximation ratios for all GNNs. We further showed that adding coloring or weak coloring to a node feature improves these approximation ratios. This indicates that preprocessing and feature engineering theoretically strengthen model capability.

**Acknowledgments**

This work was supported by JSPS KAKENHI Grant Number 15H01704. MY is supported by the JST PRESTO program JPMJPR165A.

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
