[Supplementary Material · approximation_1910241941_supp.pdf]

# A Proofs

**Lemma 11** ([24]). *If the input graph is degree-bounded and input size is bounded by a constant, each node needs to transmit and process only a constant number of bits.*

*Proof of Theorem 1.* We prove the case of $\mathcal{L} = \mathrm{VV_C}$. The proof for other cases can be done similarly. Let $\mathcal{P}_{\mathrm{GNNs}}$ be the set of graph problems that at least one $\mathrm{VV_C}$-GNN can solve and $\mathcal{P}_{\mathrm{algo}}$ be the set of graph problems that at least one distributed local algorithm on the $\mathrm{VV_C}(1)$ model can solve. Theorem 1 says that $\mathcal{P}_{\mathrm{GNNs}} = \mathcal{P}_{\mathrm{algo}}$. We now prove the following two lemmas.

**Lemma 12.** *For any $\mathrm{VV_C}$-GNN, there exists a distributed local algorithm on the $\mathrm{VV_C}(1)$ model that solves the same set of graph problems as the $\mathrm{VV_C}$-GNN.*

**Lemma 13.** *For any distributed local algorithm on the $\mathrm{VV_C}(1)$ model, there exists a $\mathrm{VV_C}$-GNN that solves the same set of graph problems as the distributed local algorithm.*

If these lemmas hold, for any $P \in \mathcal{P}_{\mathrm{GNNs}}$, there exists a $\mathrm{VV_C}$-GNN that solves $P$. From Lemma 12, there exists a distributed local algorithm on the $\mathrm{VV_C}(1)$ model that solves $P$. Therefore, $P \in \mathcal{P}_{\mathrm{algo}}$ and $\mathcal{P}_{\mathrm{GNNs}} \subseteq \mathcal{P}_{\mathrm{algo}}$. Conversely, $\mathcal{P}_{\mathrm{algo}} \subseteq \mathcal{P}_{\mathrm{GNNs}}$ holds by the same argument. Therefore, $\mathcal{P}_{\mathrm{algo}} = \mathcal{P}_{\mathrm{GNNs}}$.

Proof of Lemma 12: Let $N$ be an arbitrary $\mathrm{VV_C}$-GNN and $L$ be the number of layers of $N$. The inference of $N$ itself is a distributed local algorithm on the $\mathrm{VV_C}(1)$ model that communicates with neighboring nodes in $L$ rounds. Namely, the message from the node $v$ to its $i$-th port in the $l$-th communication round is a pair $(z_v^{(l)}, i)$, and each node calculates the next message baed on the received messages and the function $f$. Finally, each node calculates the output from the obtained embedding without communication.

Proof of Lemma 13: Let $A$ be an arbitrary distributed local algorithm and $L$ be the number of communication rounds of $A$. Let $F$ be a set of possible input features. From Assumption 3, the cardinality of $F$ is finite. Let $m_{vi}^{(l)} \in \mathbb{R}^{d_l}$ be the message that node $v$ receives from $i$-th port in the $l$-th communication round and $s_v^{(l)} \in \mathbb{R}^{d_l}$ be the internal state of node $v$ in the $l$-th communication round. $s_v^{(1)}$ is the input to node $v$ (e.g., the degree of $v$). Note that we can assume the dimensions of $m_{vi}^{(l)}$ and $s_v^{(l)}$ to be the constant $d_l$ without loss of generality by Lemma 11. Let $g_j^{(0)}(s_v^{(1)}) \colon F \to \mathbb{R}^{d_1}$ be the function that calculates the message to the $j$-th port in the first communication round from the degree information. Let $g_j^{(l)}(m_1^{(l)}, m_2^{(l)}, \ldots, m_\Delta^{(l)}, s^{(l)}) \colon \mathbb{R}^{d_l(\Delta+1)} \to \mathbb{R}^{d_{l+1}}$ be the function that calculates the message to the $j$-th port in the $(l+1)$-th communication round from the received messages and the internal state in the $l$-th communication round ($1 \le l \le L-1$). Let $g^{(l)}(m_1^{(l)}, m_2^{(l)}, \ldots, m_\Delta^{(l)}, s^{(l)}) \colon \mathbb{R}^{d_l(\Delta+1)} \to \mathbb{R}^{d_{l+1}}$ be the function that calculates the internal state in the $(l+1)$-th communication round from the received messages and the internal state in the $l$-th communication round ($1 \le l \le L-1$). Let $g^{(L)}(m_1^{(L)}, m_2^{(L)}, \ldots, m_\Delta^{(L)}, s^{(L)}) \colon \mathbb{R}^{d_L(\Delta+1)} \to Y$ be the function that determines the output from the received messages and the internal state in the $L$-th communication round. Then, we construct a $\mathrm{VV_C}$-GNN that solves the same set of graph problems as $A$. Namely, let $f^{(1)} \colon \mathbb{R}^{d_1 + (d_1+1)\Delta} \to \mathbb{R}^{d_2(\Delta+1)}$ be

$$f^{(1)}(z_v^{(1)}, z_{p_{\mathrm{tail}}(v,1)}^{(1)}, p_{\mathrm{n}}(v,1), z_{p_{\mathrm{tail}}(v,2)}^{(1)}, p_{\mathrm{n}}(v,2), \ldots, z_{p_{\mathrm{tail}}(v,\Delta)}^{(1)}, p_{\mathrm{n}}(v,\Delta)) =$$
$$\mathrm{CONCAT}(g_1^{(1)}(g_{p_{\mathrm{n}}(v,1)}^{(0)}(z_{p_{\mathrm{tail}}(v,1)}^{(1)}), g_{p_{\mathrm{n}}(v,2)}^{(0)}(z_{p_{\mathrm{tail}}(v,2)}^{(1)}), \ldots, g_{p_{\mathrm{n}}(v,\Delta)}^{(0)}(z_{p_{\mathrm{tail}}(v,\Delta)}^{(1)}), z_v^{(1)}),$$
$$g_2^{(1)}(g_{p_{\mathrm{n}}(v,1)}^{(0)}(z_{p_{\mathrm{tail}}(v,1)}^{(1)}), g_{p_{\mathrm{n}}(v,2)}^{(0)}(z_{p_{\mathrm{tail}}(v,2)}^{(1)}), \ldots, g_{p_{\mathrm{n}}(v,\Delta)}^{(0)}(z_{p_{\mathrm{tail}}(v,\Delta)}^{(1)}), z_v^{(1)}),$$
$$\ldots,$$
$$g_\Delta^{(1)}(g_{p_{\mathrm{n}}(v,1)}^{(0)}(z_{p_{\mathrm{tail}}(v,1)}^{(1)}), g_{p_{\mathrm{n}}(v,2)}^{(0)}(z_{p_{\mathrm{tail}}(v,2)}^{(1)}), \ldots, g_{p_{\mathrm{n}}(v,\Delta)}^{(0)}(z_{p_{\mathrm{tail}}(v,\Delta)}^{(1)}), z_v^{(1)}),$$
$$g^{(1)}(g_{p_{\mathrm{n}}(v,1)}^{(0)}(z_{p_{\mathrm{tail}}(v,1)}^{(1)}), g_{p_{\mathrm{n}}(v,2)}^{(0)}(z_{p_{\mathrm{tail}}(v,2)}^{(1)}), \ldots, g_{p_{\mathrm{n}}(v,\Delta)}^{(0)}(z_{p_{\mathrm{tail}}(v,\Delta)}^{(1)}), z_v^{(1)}))$$

and let $f^{(l)} \colon \mathbb{R}^{d_l(\Delta+1) + (d_l(\Delta+1)+1)\Delta} \to \mathbb{R}^{d_{l+1}(\Delta+1)}$ ($2 \le l \le L-1$) be

$$f^{(l)}(z_v^{(l)}, z_{p_{\mathrm{tail}}(v,1)}^{(l)}, p_{\mathrm{n}}(v,1), z_{p_{\mathrm{tail}}(v,2)}^{(l)}, p_{\mathrm{n}}(v,2), \ldots, z_{p_{\mathrm{tail}}(v,\Delta)}^{(l)}, p_{\mathrm{n}}(v,\Delta)) =$$
$$\mathrm{CONCAT}(g_1^{(l)}(\pi_{p_{\mathrm{n}}(v,1)}^{(l)}(z_{p_{\mathrm{tail}}(v,1)}^{(l)}), \pi_{p_{\mathrm{n}}(v,2)}^{(l)}(z_{p_{\mathrm{tail}}(v,2)}^{(l)}), \ldots, \pi_{p_{\mathrm{n}}(v,\Delta)}^{(l)}(z_{p_{\mathrm{tail}}(v,\Delta)}^{(l)}), \pi_{\Delta+1}^{(l)}(z_v^{(l)})),$$

$$g_2^{(l)}(\pi_{p_n(v,1)}^{(l)}(\boldsymbol{z}_{p_{\text{tail}}(v,1)}^{(l)}), \pi_{p_n(v,2)}^{(l)}(\boldsymbol{z}_{p_{\text{tail}}(v,2)}^{(l)}), \dots, \pi_{p_n(v,\Delta)}^{(l)}(\boldsymbol{z}_{p_{\text{tail}}(v,\Delta)}^{(l)}), \pi_{\Delta+1}^{(l)}(\boldsymbol{z}_v^{(l)})),$$

$$\dots,$$

$$g_\Delta^{(l)}(\pi_{p_n(v,1)}^{(l)}(\boldsymbol{z}_{p_{\text{tail}}(v,1)}^{(l)}), \pi_{p_n(v,2)}^{(l)}(\boldsymbol{z}_{p_{\text{tail}}(v,2)}^{(l)}), \dots, \pi_{p_n(v,\Delta)}^{(l)}(\boldsymbol{z}_{p_{\text{tail}}(v,\Delta)}^{(l)}), \pi_{\Delta+1}^{(l)}(\boldsymbol{z}_v^{(l)})),$$

$$g^{(l)}(\pi_{p_n(v,1)}^{(l)}(\boldsymbol{z}_{p_{\text{tail}}(v,1)}^{(l)}), \pi_{p_n(v,2)}^{(l)}(\boldsymbol{z}_{p_{\text{tail}}(v,2)}^{(l)}), \dots, \pi_{p_n(v,\Delta)}^{(l)}(\boldsymbol{z}_{p_{\text{tail}}(v,\Delta)}^{(l)}), \pi_{\Delta+1}^{(l)}(\boldsymbol{z}_v^{(l)}))),$$

where $\pi_i^{(l)}(h)\colon d_l(\Delta+1) \to d_l$ selects the $i$-th component from $h$ ($2 \le l \le L$, $1 \le i \le \Delta+1$), namely, $\pi_i^{(l)}(h)_j = \boldsymbol{z}_{d_l i + j}$ ($1 \le j \le d_l$). Finally, let $f^{(L)}\colon \mathbb{R}^{d_L(\Delta+1)+(d_L(\Delta+1)+1)\Delta} \to Y$ be

$$f^{(L)}(\boldsymbol{z}_v^{(L)}, \boldsymbol{z}_{p_{\text{tail}}(v,1)}^{(L)}, p_n(v,1), \boldsymbol{z}_{p_{\text{tail}}(v,2)}^{(L)}, p_n(v,2), \dots, \boldsymbol{z}_{p_{\text{tail}}(v,\Delta)}^{(L)}, p_n(v,\Delta)) =$$
$$g^{(L)}(\pi_{p_n(v,1)}^{(L)}(\boldsymbol{z}_{p_{\text{tail}}(v,1)}^{(L)}), \pi_{p_n(v,2)}^{(L)}(\boldsymbol{z}_{p_{\text{tail}}(v,2)}^{(L)}), \dots, \pi_{p_n(v,\Delta)}^{(L)}(\boldsymbol{z}_{p_{\text{tail}}(v,\Delta)}^{(L)}), \pi_{\Delta+1}^{(L)}(\boldsymbol{z}_v^{(L)}))$$

Intuitively, the embedding of the node $v$ in the $l$-th layer is the concatenation of all the messages that $v$ sends and the internal state of $v$ in the $l$-th communication round of $A$. We now prove that $\pi_{p_n(v,i)}^{(l)}(\boldsymbol{z}_{p_{\text{tail}}(v,i)}^{(l)}) = \boldsymbol{m}_{vi}^{(l)}$ and $\pi_{\Delta+1}^{(l)}(\boldsymbol{z}_v^{(l)}) = \boldsymbol{s}_v^{(l)}$ ($2 \le l \le L$) hold by induction. First, $\boldsymbol{z}_v^{(1)} = \boldsymbol{s}_v^{(1)}$ and $g_{p_n(v,i)}^{(0)}(\boldsymbol{z}_{p_{\text{tail}}(v,i)}^{(1)}) = \boldsymbol{m}_{vi}^{(1)}$ hold by definition. Therefore,

$$\pi_{p_n(v,i)}^{(2)}(\boldsymbol{z}_{p_{\text{tail}}(v,i)}^{(2)})$$
$$= g_{p_n(v,i)}^{(1)}(g_{p_n(p_{\text{tail}}(v,i),1)}^{(0)}(\boldsymbol{z}_{p_{\text{tail}}(p_{\text{tail}}(v,i),1)}^{(1)}), \dots, g_{p_n(p_{\text{tail}}(v,i),\Delta)}^{(0)}(\boldsymbol{z}_{p_{\text{tail}}(p_{\text{tail}}(v,i),\Delta)}^{(1)}), \boldsymbol{z}_{p_{\text{tail}}(v,i)}^{(1)})$$
$$= g_{p_{\text{tail}}(v,i)}^{(1)}(\boldsymbol{m}_{p_{\text{tail}}(v,i)1}^{(1)}, \boldsymbol{m}_{p_{\text{tail}}(v,i)2}^{(1)}, \dots, \boldsymbol{m}_{p_{\text{tail}}(v,i)\Delta}^{(1)}, \boldsymbol{z}_{p_{\text{tail}}(v,i)}^{(1)})$$
$$= \boldsymbol{m}_{vi}^{(2)}$$

and

$$\pi_{\Delta+1}^{(2)}(\boldsymbol{z}_v^{(2)})$$
$$= g^{(1)}(g_{p_n(v,1)}^{(0)}(\boldsymbol{z}_{p_{\text{tail}}(v,1)}^{(1)}), g_{p_n(v,2)}^{(0)}(\boldsymbol{z}_{p_{\text{tail}}(v,2)}^{(1)}), \dots, g_{p_n(v,\Delta)}^{(0)}(\boldsymbol{z}_{p_{\text{tail}}(v,\Delta)}^{(1)}), \boldsymbol{z}_v^{(1)})$$
$$= g^{(1)}(\boldsymbol{m}_{v1}^{(1)}, \boldsymbol{m}_{v2}^{(1)}, \dots, \boldsymbol{m}_{v\Delta}^{(1)}, \boldsymbol{s}_v^{(1)})$$
$$= \boldsymbol{s}_v^{(2)}$$

In the induction step, let $\pi_{p_n(v,i)}^{(k)}(\boldsymbol{z}_{p_{\text{tail}}(v,i)}^{(k)}) = \boldsymbol{m}_{vi}^{(k)}$ and $\pi_{\Delta+1}^{(k)}(\boldsymbol{z}_v^{(k)}) = \boldsymbol{s}_v^{(k)}$ hold. Then,

$$\pi_{p_n(v,i)}^{(k+1)}(\boldsymbol{z}_{p_{\text{tail}}(v,i)}^{(k+1)})$$
$$= g_{p_n(v,i)}^{(k)}(\pi_{p_n(p_{\text{tail}}(v,i),1)}^{(k)}(\boldsymbol{z}_{p_{\text{tail}}(p_{\text{tail}}(v,i),1)}^{(k)}), \dots, \pi_{p_n(p_{\text{tail}}(v,i),\Delta)}^{(k)}(\boldsymbol{z}_{p_{\text{tail}}(p_{\text{tail}}(v,i),\Delta)}^{(k)}), \pi_{\Delta+1}^{(k)}(\boldsymbol{z}_{p_{\text{tail}}(v,i)}^{(k)}))$$
$$= g_{p_n(v,i)}^{(k)}(\boldsymbol{m}_{p_{\text{tail}}(v,i)1}^{(k)}, \boldsymbol{m}_{p_{\text{tail}}(v,i)2}^{(k)}, \dots, \boldsymbol{m}_{p_{\text{tail}}(v,i)\Delta}^{(k)}, \boldsymbol{s}_{p_{\text{tail}}(v,i)}^{(k)})$$
$$= \boldsymbol{m}_{vi}^{(k+1)}$$

and

$$\pi_{\Delta+1}^{(k+1)}(\boldsymbol{z}_v^{(k+1)})$$
$$= g^{(k)}(\pi_{p_n(v,1)}^{(k)}(\boldsymbol{z}_{p_{\text{tail}}(v,1)}^{(k)}), \pi_{p_n(v,2)}^{(k)}(\boldsymbol{z}_{p_{\text{tail}}(v,2)}^{(k)}), \dots, \pi_{p_n(v,\Delta)}^{(k)}(\boldsymbol{z}_{p_{\text{tail}}(v,\Delta)}^{(k)}), \pi_{\Delta+1}^{(k)}(\boldsymbol{z}_v^{(k)}))$$
$$= g^{(k)}(\boldsymbol{m}_{v1}^{(k)}, \boldsymbol{m}_{v2}^{(k)}, \dots, \boldsymbol{m}_{v\Delta}^{(k)}, \boldsymbol{s}_v^{(k)})$$
$$= \boldsymbol{s}_v^{(k+1)}$$

By induction, $\pi_{p_n(v,i)}^{(l)}(\boldsymbol{z}_{p_{\text{tail}}(v,i)}^{(l)}) = \boldsymbol{m}_{vi}^{(l)}$ and $\pi_{\Delta+1}^{(l)}(\boldsymbol{z}_v^{(l)}) = \boldsymbol{s}_v^{(l)}$ ($2 \le l \le L$) hold. Therefore, the final output of this $VV_C$-GNN is the same as that of $A$. $\qquad\square$

**Lemma 14** ([10]). *Let $\mathcal{P}_{SB(1)}$, $\mathcal{P}_{MB(1)}$, and $\mathcal{P}_{VV_C(1)}$ be the set of graph problems that distributed local algorithms on SB(1), MB(1), and $VV_C(1)$ models can solve only with the degree features, respectively. Then, $\mathcal{P}_{SB(1)} \subsetneq \mathcal{P}_{MB(1)} \subsetneq \mathcal{P}_{VV_C(1)}$.*

*Proof of Theorem 2.* From Theorem 1 and Lemma 14, $\mathcal{P}_{\text{SB}(1)} = \mathcal{P}_{\text{SB-GNNs}} \subsetneq \mathcal{P}_{\text{MB}(1)} = \mathcal{P}_{\text{MB-GNNs}} \subsetneq \mathcal{P}_{\text{VV}_{\text{C}}(1)} = \mathcal{P}_{\text{VV}_{\text{C}}\text{-GNNs}}$ holds. □

**Lemma 15** ([1, 24]). *Let $A$ be any distributed local algorithm with $L$ communication rounds, $G = (V, E)$ and $G' = (V', E')$ be any graphs, $p$ and $p'$ be any port numberings of $G$ and $G'$, $\boldsymbol{X}$ and $\boldsymbol{X}'$ be any input to the nodes $V$ and $V'$, and $v$ and $v'$ be any nodes of $G$ and $G'$, respectively. If the radius-$L$ local views of $v$ and $v'$ are the same, the outputs of $A$ for $v$ and $v'$ are the same.*

*Proof of Theorem 3.* $\mathcal{P}_{\text{CPNGNNs}} \subseteq \mathcal{P}_{\text{VV}_{\text{C}}\text{-GNNs}}$ clearly holds because any CPNGNN is a VV$_\text{C}$-GNN. Now, we prove $\mathcal{P}_{\text{CPNGNNs}} \supseteq \mathcal{P}_{\text{VV}_{\text{C}}\text{-GNNs}}$. We decompose CPNGNNs into two parts. The first part $\Phi_\theta$ corresponds to lines 3-8 of in Algorithm 2 (i.e., communication round) and the second part $\Psi_{\theta'}$ corresponds to the tenth line of Algorithm 2 (i.e., calculating the final embedding). Namely, $\Phi_\theta(G, \boldsymbol{X}, v) = \boldsymbol{z}_v^{(L+1)}$ and $\Psi_{\theta'}(\boldsymbol{z}_v^{(L+1)}) = \boldsymbol{z}_v$, where $\theta$ and $\theta'$ are parameters of the network (i.e., $\boldsymbol{W}^{(l)}$ ($l = 1, 2, \ldots, L$) and the parameters of MLP).

Let $\boldsymbol{W}^{(1)}, \boldsymbol{W}^{(2)}, \ldots, \boldsymbol{W}^{(L)}$ be the identity matrices. Let $G = (V, E)$ and $G' = (V, E)$ be any graphs, $p$ and $p'$ be any port numberings of $G$ and $G'$, $\boldsymbol{X}$ and $\boldsymbol{X}'$ be input vectors whose elements are non-negative integers, and $v$ and $v'$ be any nodes of $G$ and $G'$, respectively.

**Lemma 16.** *If the radius-$L$ local views of $v$ and $v'$ are the same, $\Phi_\theta(G, \boldsymbol{X}, v) = \Phi_\theta(G', \boldsymbol{X}', v')$.*

*Proof of Lemma 16.* We prove that for any $v \in V$, we can reconstruct the radius-$l$ local view of $v$ from $\boldsymbol{z}_v^{(l+1)}$ using mathematical induction. When $l = 1$, $\boldsymbol{z}_v^{(2)} = \text{CONCAT}(\boldsymbol{z}_v^{(1)}, \boldsymbol{z}_{p_{\text{tail}}(v,1)}^{(1)}, p_\text{n}(v, 1), \boldsymbol{z}_{p_{\text{tail}}(v,2)}^{(1)}, p_\text{n}(v, 2), \ldots, \boldsymbol{z}_{p_{\text{tail}}(v,\Delta)}^{(1)}, p_\text{n}(v, \Delta))$. We omit the ReLU function because the vector is always non-negative. The input vector of node $v$ is $\boldsymbol{z}_v^{(1)}$. The input vector of the node that sends the message to the $i$-th port of node $v$ is $\boldsymbol{z}_{p_{\text{tail}}(v,i)}^{(1)}$, and its port number that sends to the node $v$ is $p_\text{n}(v, i)$. Therefore, $\boldsymbol{z}_v^{(2)}$ includes sufficient information on the input vector of node $v$, input vectors of neighboring nodes, and port numbering of the incident edges. In the induction step, for any $v \in V$, $\boldsymbol{z}_v^{(k+1)}$ contains sufficient information to reconstruct the radius-$k$ local view of $v$. When $l = k + 1$, $\boldsymbol{z}_v^{(k+2)} = \text{CONCAT}(\boldsymbol{z}_v^{(k+1)}, \boldsymbol{z}_{p_{\text{tail}}(v,1)}^{(k+1)}, p_\text{n}(v, 1), \boldsymbol{z}_{p_{\text{tail}}(v,2)}^{(k+1)}, p_\text{n}(v, 2), \ldots, \boldsymbol{z}_{p_{\text{tail}}(v,\Delta)}^{(k+1)}, p_\text{n}(v, \Delta))$. From the inductive hypothesis, we can reconstruct the radius-$k$ local view $\mathcal{T}_v$ of node $v$. For any $i$, we can reconstruct the radius-$k$ local view $\mathcal{T}_i$ of the node that sends a message to the $i$-th port of the node $v$. We call this node $u_i$ for the purpose of explanation. Note that we cannot identify which node $u$ is. We merge all of $\mathcal{T}_i$ with $\mathcal{T}_v$ to construct the radius-$(k+1)$ local view of node $v$. There exists at least one child of the root of $\mathcal{T}_i$ that is compatible when we merge $\mathcal{T}_i$ and $\mathcal{T}_v$ because $v$ is an adjacent node of $u_i$. In other words, there exists a child $c$ of the root of $\mathcal{T}_i$ such that the port numbering between $c$ and $u$ is the same as that between $v$ and $u$ and the subtree of $\mathcal{T}_i$ where the root is $c$ is the same as the radius-$(k-1)$ local view of $v$ without the subtree where the root is $v$. The node $c$ corresponds to node $v$. Note that $c$ may not be $v$ itself, but this is irrelevant because the resulting tree is isomorphic. After we merge all $\mathcal{T}_i$, the resulting tree is the radius-$(k+1)$ local view of $v$. By mathematical induction, for any $v \in V$, we can reconstruct the radius-$l$ local view of $v$ from $\boldsymbol{z}_v^{(l+1)}$. Therefore, if the radius-$L$ local views of $v$ and $v'$ are the same, the outputs $\boldsymbol{z}_v^{(L+1)}$ and $\boldsymbol{z}_{v'}^{(L+1)}$ must be the same. □

Furthermore, if the input vectors $\boldsymbol{X}$ are bounded non-negative integers (i.e., $\boldsymbol{X} \in (\mathbb{N} \cap [0, \alpha])^{n \times d_1}$ for some $\alpha \in \mathbb{N}$), the output vector $\Phi_\theta(G, \boldsymbol{X}, v)$ consists of bounded non-negative integers (i.e., $\Phi_\theta(G, \boldsymbol{X}, v) \in (\mathbb{N} \cap [0, \beta])^{d_{L+1}}$ for some $\beta \in \mathbb{N}$). Let $N$ be any VV$_\text{C}$-GNN. From Lemmas 12, there exists a distributed local algorithm $A$ that solves the same set of graph problems as $N$. Let $f(G, \boldsymbol{X}, v) \in \{0, 1\}^{|Y|}$ represent the one-hot vector of the output of $A$. From Lemma 15 and 16, there exists a function $h(\boldsymbol{v}) \colon (\mathbb{N} \cap [0, \beta])^{d_{L+1}} \to \{0, 1\}^{|Y|}$ such that $h \circ \Phi_\theta(G, \boldsymbol{X}, v) = f(G, \boldsymbol{X}, v)$. Let $h' \colon [0, \beta]^{d_{L+1}} \to [0, 1]^{|Y|}$ be a linear interpolation of $h$. Because $h'$ is continuous and bounded, from the universal approximation theorem [5], there exists a parameter $\theta'$ such that for any $\boldsymbol{v} \in [0, \beta]^{d_{L+1}}$, $\|\Psi_{\theta'}(\boldsymbol{v}) - h'(\boldsymbol{v})\|_2 < 1/3$. Therefore, the maximum index of $\Psi_{\theta'}(\boldsymbol{z}_v^{(L+1)})$ is the same as that of $h(\boldsymbol{z}_v^{(L+1)})$ and the output of this network is the same as that of $N$ for any input. □

---

**Algorithm 3** Calculating a consistent port numbering

---

**Require:** Graph $G = (V, E)$.
**Ensure:** Consistent port numbering $p$.
 1: $c_v \leftarrow 0 \ \forall v \in V$
 2: $p \leftarrow$ empty dictionary
 3: **for** $\{u, v\} \in E$ **do**
 4:    $c_u \leftarrow c_u + 1$
 5:    $c_v \leftarrow c_v + 1$
 6:    $p((u, c[u])) = (v, c[v])$
 7:    $p((v, c[v])) = (u, c[u])$
 8: **end for**
 9: **return** $p$

---

**Lemma 17** ([3, 6, 15]). *The optimal approximation ratio of the VV$_C$ model for the minimum dominating set problem is $\Delta + 1$.*

**Lemma 18** ([3]). *If inputs contain weak 2-coloring, the optimal approximation ratio of the VV$_C$ model for the minimum dominating set problem is $\frac{\Delta+1}{2}$.*

**Lemma 19** ([3]). *If inputs contain 2-coloring, the optimal approximation ratio of the VV$_C$ model for the minimum dominating set problem is $\frac{\Delta+1}{2}$.*

**Lemma 20** ([2, 6, 15]). *The optimal approximation ratio of the VV$_C$ model for the minimum vertex cover problem is $2$.*

**Lemma 21** ([3, 6]). *The optimal approximation ratio of the VV$_C$ model for the maximum matching problem does not exist.*

**Lemma 22** ([3]). *If inputs contain weak 2-coloring, the optimal approximation ratio of the VV$_C$ model for the maximum matching problem is $\frac{\Delta+1}{2}$.*

**Lemma 23** ([3]). *For any $\Delta \geq 1$ and $\varepsilon > 0$, there is a distributed local algorithm on the VV$_C$ model with approximation ratio factor $1 + \varepsilon$ for maximum matching in 2-colored graphs.*

Theorems 4, 5, 6, 7, 8, 9, and 10 immediately follow from Lemmas 17, 18, 19, 20, 21, 22, and 23, respectively, because from Theorems 1 and 3, the set of graph problems that CPNGNNs can solve is the same as that that the VV$_C$ model can.

## B  How to Calculate a Consistent Port Numbering and a Weak 2-Coloring

A consistent port numbering can be calculated in linear time. We show the pseudo code in Algorithm 3. A weak 2-coloring can be also calculated in linear time by breadth first search. We show the pseudo code in Algorithm 4. Note that if the input graph is bipartite, Algorithm 4 returns a 2-coloring of the input graph.

## C  Experiments

In this section, we confirm that CPNGNNs can solve a graph problem that existing GNNs cannot through experiments. We use a toy task named finding single leaf [10]. In this problem, the input is a star graph, and the output must be a single leaf of the graph. If the input graph is not a star graph, GNNs may output any subset of nodes. Formally, this graph problem is expressed as follows:

$$\Pi(G) = \begin{cases} \{\{v\} \mid v \in V, \deg(v) = 1\} & \text{if } G \text{ is a star graph} \\ 2^V \ (\text{i.e., any subset of } V) & \text{otherwise} \end{cases}.$$

No MB-GNN can solve this problem because for any layer, the latent vector in each leaf node is identical and MB-GNNs must output the same decision for all leaf nodes.

In this experiment, we use a star graph with four nodes: one center node and three leaves used for both training and testing. We use a two-layer CPNGNN that learns the stochastic policy of node

**Algorithm 4** Calculating a weak 2-coloring
___
**Require:** Graph $G = (V, E)$.
**Ensure:** Weak 2-coloring $c$.
  1:  $f_v \leftarrow$ **false** $\forall v \in V$
  2:  $q \leftarrow$ empty queue
  3:  $v_0 \leftarrow$ an arbitrary node in $G$
  4:  $q$.push$((v_0, 0))$
  5:  $f_{v_0} \leftarrow$ **true**
  6:  **while** $q$ is not empty **do**
  7:     $(v, x) \leftarrow q$.front()
  8:     $q$.pop()
  9:     $c(v) = x$
10:     **for** $u \in \mathcal{N}(v)$ **do**
11:       **if not** $f_u$ **then**
12:         $q$.push$((u, 1 - x))$
13:         $f_u \leftarrow$ **true**
14:       **end if**
15:     **end for**
16:  **end while**
17:  **return** $c$
___

selection and train the model using the REINFORCE algorithm [29]. If the output selects only one leaf, the reward is 1, and otherwise, the reward is $-1$. We ran 10 trials with different seeds. After 10000 iterations of training, the model solves the finding single leaf problem in all trials. However, we train GCN [12], GraphSAGE [9], and GAT [26] to solve this task, but none of them could solve the finding single leaf problem, as our theory shows. This indicates that the existing GNNs cannot solve such a simple combinatorial problem whereas out proposed model can.