[Reviews · NeurIPS 2019]

Reviewer 1



The article contains 10 theorems which constitute the backbone of the paper. In my opinion, the most important is Theorem 1, which links GNNs and distributed local algorithms. Later approximation results for GNNs are dependent on earlier results obtained for distributed local algorithms. The key technical part of the proof is presented in Lemma 13 in the Appendix that states that every distributed local algorithm can be simulated using a GNN. The novel GNN models (VV_C-GNNs and its subclass CPNGNNs) proposed in the paper can be characterized by the following high-level description: proposed classes of GNNs "can send different messages to neighboring nodes by using port numbering, and this strengthens model capability". In the following theorems, a number of lower and upper approximation bounds are provided. Example: Theorem 7. The optimal approximation ratio of CPNGNNs for the minimum vertex cover problem is 2. In other words, there exists a CPNGNN that can solve 2-approximation for the minimum vertex and for any 1 ≤ α < 2, there exists no CPNGNN that can solve α-approximation for the minimum vertex cover problem. ============== Update: I appreciate the extra information on the availability of examples provided in your response. In my opinion, the choice of Reinforce is quite problematic, but the design of this experiment does not influence my assessment of the paper. I maintain my score.

Reviewer 2



The paper proposed a new family of GNNs that are more powerful than the existing ones (e.g., GCN, GAT, GraphSAGE). The notion of powerfulness is defined as the capability for model instances in a model class in solving combinatorial tasks over graphs. The key idea is to leverage port numbering of the graphs, and to take advantage of node coloring features (which seems necessary in order to go beyond trivial approximation ratios). The authors then analyzed the approximation ratio of the proposed CPNGNNs over several canonical combinatorial tasks. Overall, the paper is well written and technically sound, and has offered a unified view of several seminal GNN model classes. The work can be viewed as an extension of Xu et al., 2018 [1] (in which the graph isomorphism task was the only focus). Major concerns: 1. None of the theorems are backed up by empirical results. I believe experiments are crucial to verify the correctness of the theorems, and also to demonstrate the usefulness of CPNGNNs in practice. The authors would probably benefit from having at least one of the following: (1) CPNGNNs lead to better approximation ratio than existing machine learning solvers (e.g. methods in Bello et al., 2016, Vinyals et al., 2015) on synthetic graph combinatorial tasks analyzed in sect 6.2; and (2) CPNGNNs leads to improved/competitive performance on real-world benchmarks as compared to GIN, GCN, GraphSAGE etc. 2. As the authors pointed out, approximation ratios given by Theorem 4 and Theorem 8 are trivial. The improved ratio, i.e., (max_degree + 1) / 2 obtained by leveraging additional graph coloring features, also seems disappointedly loose. While it has been proven in the paper that no GNNs can do better, the paper can be strengthened if the authors could provide more insights/explanations for those unsatisfactory ratios. 3. Definition of "solving" a given combinatorial task seems tricky (L121-122). If my understanding is correct, a GNN model class is considered to be able to solve the task as long as it contains a single model instance that solves the task. However, there doesn’t seem to be a straightforward mechanism to identify the right model instance in the given model class. Minor questions: * Would k-coloring (k > 2) make any difference in the approximation ratio? [1] Xu, Keyulu, et al. "How powerful are graph neural networks?." ICLR 2019.

Reviewer 3



Originality: The proposed classes are new but are minor modifications of existing ones. Quality: Technically sound. Clarity: It was easy to follow the argument. But the authors must be clear in the statement of theorems. For example, in Theorem 7, can we use a single choice of parameters to achieve 2-approximation or we have to change parameters depending on the input graph? Significance: As I mentioned above, I'm not sure the main results are very interesting. Why do we want to identify the best approximation ratio we can obtain with a DNN when we know that it won't be better than those of known approximation algorithms? Also, it is not clear how to learn parameters to achieve the claimed approximation ratio. Line 8: As "GNN" is not a well-defined term, it does not make much sense to say "no GNN can perform better than ..."

[Author Response · NeurIPS 2019]

Thank you all for your reviews and constructive comments. We will revise the manuscript based on your suggestions.

**Reviewer #1:** ▶ *Add more examples showing that the new GNNs are more expressive than previously considered classes of GNNs.* The finding single leaf problem is the only known problem that does not belong to $\mathcal{P}_{\text{VV}_{\text{C}}\text{-GNNs}} \backslash \mathcal{P}_{\text{MB-GNNs}}$ . It has been a long-standing open problem to find other such problems in the field of distributed local algorithms [10]. If this open problem is solved in the distributed algorithm community in the future, we can give an example thanks to Theorem 1. It should be noted that the approximation of the minimum vertex cover problem provably belongs to $\mathcal{P}_{\text{VV}_{\text{C}}\text{-GNNs}}$ (Theorem 7) whereas it is not known whether this problem belongs to $\mathcal{P}_{\text{MB-GNNs}}$ or not. ▶ *Another extension of GNNs was proposed in https://arxiv.org/abs/1810.02244 - it would be interesting to compare these two approaches ...* As you pointed out, their approach is orthogonal to ours. For example, k-GNNs cannot solve the finding single leaf problem (Line 229) whereas ours can. Therefore, we can make k-GNNs more powerful using port numbering. Examining the expressive power of k-GNNs with port numbering more precisely is an interesting future work. ▶ *Why authors have selected the Reinforce algorithm for training?* We followed the existing work [4].

**Reviewer #2:** Important results in this paper are the inapproximability results (e.g., Theorem 4 and 8) rather than the approximability results. The best approximation ratios that GNNs can achieve are far worse than many researchers considered. Moreover, as Reviewer #1 pointed out, the most important contribution is to show a link between GNNs and distributed local algorithms (Theorem 1). These surprising results must have a large impact on the NeurIPS community. ▶ *Definition of "solving" a given combinatorial task seems tricky (L121-122). If my understanding is correct, a GNN model class is considered to be able to solve the task as long as it contains a single model instance that solves the task.* As you pointed out, the definition of solvability is fairly loose in this paper. Therefore, the inapproximability results become extremely strong. It indicates that there exist no model instances that can solve these graph problems, and any elaborated training procedures cannot find any model instance that solve these problems. ▶ *the paper can be strengthened if the authors could provide more insights/explanations for those unsatisfactory ratios.* We show an illustrative example of the minimum dominating set problem in Figure 1. ▶ *I believe experiments are crucial to verify the correctness of the theorems...* We gave a mathematical proof for each theorem, which verifies the correctness of the theorem more rigorously than any empirical experiments.

Figure 1: **Minimum Dominating Set Problem:** GNNs output invalid or redundant solutions without coloring because the input graph is symmetrical. With coloring, GNNs can distinguish adjacent nodes, but cannot identify the global structure. Thus GNNs output suboptimal solutions.

**Reviewer #3:** ▶ *a DNN runs in polynomial time and we have inapproximability results for polynomial-time algorithms, we already know that it cannot beat known approximation algorithms in terms of approximation ratio.* We showed the approximation ratios of GNNs are far worse than known inapproximablity results for polynomial time algorithms. For example, there exists a $(\mathcal{H}_{\Delta+1} - \frac{1}{2})$-approximation algorithm for the minimum dominating set problem [A], where $\mathcal{H}_i$ is the $i$-th harmonic number. Considering $\mathcal{H}_{\Delta+1} = O(\log \Delta)$, the best approximation ratio $(\Delta + 1)$ of GNNs is far worse than this algorithm. Moreover, GNNs cannot solve even an easy instance as Figure 1 shows. This fact has been overlooked in the GNN community. ▶ *I guess the reason why people try to use DNNs for combinatorial problems is its empirical performance. ... Why do we want to identify the best approximation ratio we can obtain with a DNN when we know that it won't be better than those of known approximation algorithms?* Indeed, GNNs are popular for its empirical performance. However, we consider providing a theoretical guarantee is also important. For example, when one determines the schedule of product releases using a combinatorial solver without any theoretical guarantee, it may output a far worse solution than the optimal solution and causes an enormous loss. We proved GNN cannot use such applications that need a theoretical guarantee. ▶ *in Theorem 7, can we use a single choice of parameters to achieve 2-approximation or we have to change parameters depending on the input graph?* In all theorems, we use a single choice of parameters to achieve the approximation ratios (see Line 114 and 121). ▶ *Line 8: As "GNN" is not a well-defined term, it does not make much sense to say "no GNN can perform better than ..."* We intended GNN meant MB-GNN, which include most of GNNs in the literature (Line 155 - 161). We will clarify it.

**References:** [A] Miroslav Chlebík and Janka Chlebíková. Approximation hardness of dominating set problems in bounded degree graphs. *Inf. Comput.,* 2008.

[Meta-Review · NeurIPS 2019]

This paper studies the approximation power of GNNs on combinatorial optimization problems. Its main result is to establish an equivalence between these neural networks and distributed local algorithms, and to provide a stronger family of GNNs that result in approximation algorithms for several NP-hard problems with provably better approximation ratio as alternatives. Reviewers had mixed impressions on this paper. Whereas they recognize the theoretical contribution of this work, they were also somewhat concerned by the lack of numerical examples that illustrate how the ideas developed in the theorems may apply outside the set of three NP-hard optimization examples. The AC also shares these concerns, but ultimately thinks the positive aspects of the paper outweight the negatives.